# Antioxidant, Biochemical, and In-Life Effects of *Punica granatum* L. Natural Juice vs. Clarified Juice by Polyvinylidene Fluoride Membrane

**DOI:** 10.3390/foods9020242

**Published:** 2020-02-24

**Authors:** Valeria Maria Morittu, Vincenzo Mastellone, Rosa Tundis, Monica Rosa Loizzo, Raffaella Tudisco, Alberto Figoli, Alfredo Cassano, Nadia Musco, Domenico Britti, Federico Infascelli, Pietro Lombardi

**Affiliations:** 1C.I.S.—Interdepartmental Services Centre of Veterinary for Human and Animal Health, Magna Graecia University of Catanzaro, Viale Europa, Loc. Germaneto, 88100 Catanzaro, Italy; morittu@unicz.it (V.M.M.); britti@unicz.it (D.B.); 2Department of Veterinary Medicine and Animal Production, University of Napoli Federico II, 80100 Napoli, Italy; vincenzo.mastellone@unina.it (V.M.); tudisco@unina.it (R.T.); federico.infascelli@unina.it (F.I.); pietro.lombardi@unina.it (P.L.); 3Department of Pharmacy, Health and Nutritional Sciences, University of Calabria, Via P. Bucci, Ed. Polifunzionale, 87030 Rende, Italy; tundis@unical.it (R.T.); monica_rosa.loizzo@unical.it (M.R.L.); 4Institute on Membrane Technology, Italian National Research Council (ITM-CNR), Via P. Bucci, 87036 Rende, Italy; a.figoli@itm.cnr.it (A.F.); a.cassano@itm.cnr.it (A.C.)

**Keywords:** pomegranate, clarified juice, antioxidants, hypoglycemic, functional drink

## Abstract

A clarification method was proposed to ameliorate the technological quality of fruit juices by preserving bioactive compounds. This study evaluated the in vitro antioxidant and hypoglycemic activities and the in vivo effects of *Punica granatum* L. natural (NJ) and clarified (CJ) juice by polyvinylidene fluoride (PVDF) hollow fiber membrane. CJ was more active as an antioxidant and as a α-glucosidase inhibitor than NJ. Mice were orally gavaged with water (Control), NJ, and CJ for 28 days. NJ group showed significant decrease of alanine aminotransferase, aspartate amino transferase, and creatine-phosphokinase. CJ administration was associated with urea, creatine-phosphokinase, and triglycerides values significantly lower with respect to the control. Oxidative status was ameliorated with CJ administration, showing a reactive oxygen metabolites (d-ROMs) reduction of 32% and a biological antioxidant potential (BAP) boosting of 23% compared to the control, whereas NJ did not show a similar effect. Results confirmed the beneficial properties of pomegranate juice, showing that membrane clarification may enhance such effects in terms of antioxidant activity.

## 1. Introduction

*Punica granatum* L. (pomegranate) is a shrub bush or small tree, deciduous, up to 6–7 m, densely branched, with often thorny branches widely grown in Asia, the Mediterranean border, and the American continent. Its cultivation has increased considerably in Italy in the last five years, mainly in the South, growing from 50 ha in 2012 to 1500 ha in 2017, with a current total production of about 6000 tons of fruits per year [1]. Pomegranate and its constituents have safely been consumed for centuries without any side effects. Studies of pomegranate constituents in animals showed no toxic effects, at concentrations commonly used in folk and traditional medicine [2].

In recent years, this traditional use has received attention from the scientific community; the in vitro and in vivo studies carried out demonstrated anti-inflammatory, cardio-preventive, anticancer, and antioxidant properties of pomegranate fruits [3,4]. In particular, Al-Jarallah et al. [5] showed that pomegranate extract supplementation substantially reduced levels of oxidative stress in coronary arteries and atherosclerotic plaques in SR-BI/apoE dKO mice. Additionally, ethanolic pomegranate extracts up to a concentration of 5 mg/mL promoted longevity, formation, and fertility of new generations and growth properties of *Caenorhabditis elegans* [6].

The nutritional value of pomegranate juice is most related to the fruit, a source of carbohydrates, minerals, crude fibers, and various biologically active compounds, such as vitamin C, and phenolic compounds, including punicalagin, ellagic acid, gallotannins, and anthocyanins [7].

The pericarp, the inner lamella, and the consumed part constitutes 38%, 10% and 52% of the total weight of the fruit, respectively. The juice is 78% of the portion used (45–61% of total weight), and the seeds are 22% [8]. Interestingly, the above nutraceutical properties are not limited to the edible part of pomegranate fruit; the nonedible fractions of fruit (i.e., peel and seeds), although considered as waste, also contain even higher amounts of specific nutritionally valuable and biologically active components, as compared to the edible fruit [9]. By contrast, the presence of large particles in the juice causes a turbid appearance and additional problems in the concentration process, such as off flavors due to their burning on the evaporator walls. In addition, the bioaccessibility and bioavailability of each compound differ greatly, and the most abundant antioxidants in ingested fruit are not necessarily those leading to the highest concentration of active metabolites in target tissues [10]. Fruit antioxidants are commonly mixed with different macromolecules, such as carbohydrates, lipids, and proteins, to form a food matrix that can interfere on their absorption and/or use.

In the last years, some membrane processes, such as microfiltration (MF) and ultrafiltration (UF), have been largely investigated in fruit juice processing as an alternative to conventional clarification processes based on the use of fining agents (i.e., gelatin, bentonite, polyvinylpyrrolidone, etc.) and other filtration aids [11,12,13]. These processes do not involve phase change or chemical agents and can be operated at low temperatures, preserving freshness, aroma, and nutritional values of fresh juices. Additional advantages are in terms of increased juice yield, possibility of operating in a single step, reducing working times, easy cleaning and maintenance of the equipment, reduction of waste products, and elimination of needs for pasteurization [14].

Several in vitro tests have been performed to assess the antioxidant properties of the clarified juice [15,16,17,18,19] in order to optimize the content in bioactive compounds by minimizing possible interferences due to other constituents. In particular, in our previous study [18], the chemical profile and in vitro antioxidant properties of pomegranate-clarified juice have been analyzed and compared by using polyvinylidene fluoride (PVDF) and polysulfone (PSU) hollow fiber (HF) membranes. A lower retention towards healthy constituents, particularly flavonoids and anthocyanins, in comparison to the PSU membrane was demonstrated for the PVDF membrane. Accordingly, the PVDF-clarified juice showed a greater antioxidant activity than the PSU-clarified juice.

The aim of this study was to evaluate the in vitro antioxidant and hypoglycemic activity and the effects of both natural and HF PVDF membranes-clarified juice in mice on in-life evaluations (feed and water consumption, body weight (BW), average daily gain (ADG)); biochemical parameters and oxidative status. To this purpose, the untreated (natural) and clarified juices in a ratio of 500 mg/kg BW were administered daily to Hsd:CDR male mice by oral gavage for 28 days. Few data are reported in literature on the use of *P. granatum* juice in healthy mice. For this reason, the choice of dosage to be administered was based on a quantity acceptable to drink daily for a medium-term period and corresponding to about 25 mL of juice for a person of 50 kg of BW.

## 2. Materials and Methods

### 2.1. Chemicals and Reagents

Solvay advanced polymers kindly provided PVDF Solef^®^ 6010. BASF supplied polyvinylpyrrolidone (PVP) Luviskol K-17. All the other chemicals were of analytical grade and used without further purification. N,N-dimethylformamide (DMF), ethanol, methanol, dimethyl sulfoxide (DMSO), potassium iodide, and sodium hydroxide were purchased from VWR International s.r.l. (Milan, Italy). Acarbose from Actinoplanes sp. was obtained from Serva (Heidelberg, Germany). Sodium hypochlorite, Folin-Ciocalteu reagent, β-carotene, butylated hydroxytoluene (BHT), quercetin, cyanidin 3-glucoside, ascorbic acid, chlorogenic acid, and aluminium chloride were obtained from Sigma-Aldrich S.p.a. (Milan, Italy).

### 2.2. Membrane Preparation

Hollow fiber (HF) membranes were prepared by a nonsolvent induced phase separation technique, as reported elsewhere [18]. The produced membranes were assembled in glass modules with a length of 20 cm and fixed at each extremity with epoxy glue. Each module was equipped with three hollow fibers.

### 2.3. P. granatum Juice Preparation

*P. granatum* fruits were purchased in October 2017 from a local market in Cosenza (Southern Italy). Fruits were washed and manually cut up. The juice was extracted by using an electric squeezer and prefiltered using a mesh filter, obtaining a red deep color extract. After extraction, the juice was stored at −17 °C and defrosted to room temperature before the clarification step.

### 2.4. Clarification of P. granatum Juice

*P. granatum* (pomegranate) juice was clarified by using a bench laboratory plant consisting of a stainless steel feed tank, a magnetic drive gear pump, two pressure gauges located at the inlet and outlet of the membrane module, a pressure control valve, and a multi-tube heat exchanger fed with tap water. The temperature of the feed was controlled by circulating cooling water through the heat exchanger; the axial feed flowrate and the transmembrane pressure were controlled by using a needle concentrate valve and by setting the speed of the pump. The plant was equipped with a membrane module prepared by embedding three HF membranes inside a 20-cm-long glass tube (effective membrane length 18 cm and effective membrane area 27.6 cm^2^) with epoxy resin. The juice was clarified according to the batch concentration mode recycling continuously the retentate in the feed tank and collecting separately the permeate stream. The clarification process was operated at a temperature of 25 °C, with a feed flowrate of 30 L/h and a transmembrane pressure of 0.6 bar. After juice processing, membranes were washed with distilled water at 40 °C for 30 min and cleaned with a 1% *w/w* P3 Ultrasil 53 solution (a neutral enzymatic powder detergent containing a combination of organic and inorganic surfactants, from Henkel KGaA, Dusseldorf, Germany) at 40 °C for 60 min. Finally, distilled water was flushed through the fibers for 20 min at room temperature. This cleaning protocol allowed recovering about 90–95% of the initial water permeability of the membranes. The clarified juice was stored at −17 °C until analysis.

### 2.5. Determination of Suspended Solids and Soluble Solids and pH

The suspended solids content was evaluated by centrifuging juices for 20 min at 2000 rpm. The weight of suspended solids was then registered after the removal of the supernatant and expressed in relation of the total juice (wt %). The soluble solids content was measured with a refractometer (Atago Co., Ltd., Kumamoto, Japan) at 25 °C. Data are expressed as °Brix. A digital pH meter (PC 2700, Eutech Instruments, Landsmeer, The Netherlands) was used to measure pH.

### 2.6. Total Phenols, Anthocyanins, Flavonoids, and Ascorbic Acid Content

The total phenols content was determined using the Folin-Ciocalteu method as reported by Loizzo et al. [20]. A solution of Folin-Ciocalteu reagent, sodium carbonate (15%), and juice was incubated at room temperature for 2 h. Then, the absorbance was read at 765 nm. The total phenols content was expressed as mg of chlorogenic acid equivalents/L.

The total monomeric anthocyanins content of pomegranate juices was evaluated, as previously described [20]. Concisely, 0.5 mL of juice was mixed with (a) 3.5 mL of potassium chloride buffer (0.025 M, pH 1) and (b) 3.5 mL of sodium acetate buffer (0.025 M, pH 4.5). After 15 min, the absorbance of each solution was measured at 510 and 700 nm. Results were expressed as mg of cyanidin 3-glucoside equivalents/L. The total flavonoids content was determined by using a spectrophotometric assay [20]. In brief, sodium nitrite (0.3 ml 5% *w/v*) was added to a juice solution. After 5 min, 0.6 mL of aluminum chloride (10% *w/v*) was added. At 6 min, 2 mL of sodium hydroxide (1 M) and 2.1 mL of water were also added to the mixture that was left for 15 min at room temperature. The absorbance was measured at 510 nm, and the total flavonoids content was expressed as mg of quercetin equivalents/L. The content of ascorbic acid was also determined in line with the method previously published by Klein and Perry [21]. Results were expressed as mg/100 mL of juice.

### 2.7. In Vitro Antioxidant Activity

Three assays, namely 2,2-diphenil-1-picrylhydrazyl (DPPH) radicals scavenging, Ferric-reducing antioxidant power (FRAP), and β-carotene bleaching tests, were used to investigate in vitro the antioxidant potential of pomegranate juices [20]. In a DPPH test, a mixture of DPPH (0.25 mM) and juices (5–1000 µg/mL) was prepared and was left for 30 min at room temperature. Then, the absorbance was read at 517 nm. The positive control was ascorbic acid. The FRAP test is based on the redox reaction that involves the TPTZ (2,4,6-tripyridyl-s-triazine)-Fe^3+^ complex. Concisely, the reduction of this complex to Fe^2+^/TPTZ can be examined by measuring the change in absorption at 595 nm. FRAP value is the ratio between the slope of the linear plot for reducing the Fe^3+^–TPTZ reagent by samples compared to the slope of the plot for FeSO_4_. This change is related to the reducing power of electron-donating antioxidant compounds present in the analyzed mixture. Pomegranate juices were tested at the concentration of 2.5 mg/mL, and butylated hydroxytoluene (BHT) was used as a positive control. The β-carotene bleaching test is used to investigate the potential ability of pomegranate juices to inhibit lipids peroxidation. In brief, β-carotene (1 mL, 0.2 mg/mL) was added to linoleic acid (0.02 mL) and Tween 20 (0.2 mL). The solvent was evaporated, and water was added. Then, 5 mL of the formed emulsion was transferred into tubes that contain 0.2 mL of samples at different concentrations (0.5–100 μg/mL). The absorbance was measured at 470 nm. The positive control was propyl gallate.

### 2.8. In Vitro Hypoglycaemic Properties

The hypoglycemic activity of pomegranate juices was investigated by using α-amylase and α-glucosidase inhibitory activity tests [22,23]. In the α-amylase inhibitory test, a starch solution was prepared by stirring for 15 min at 65 °C potato starch (0.125 g) in 25 mL of sodium phosphate buffer (20 mM) with sodium chloride (6.7 mM). The α-amylase solution was prepared by mixing 0.0253 g of α-amylase in 100 mL of cold water. The colorimetric reagent was prepared mixing a sodium potassium tartrate solution (12.0 g of sodium potassium tartrate in 8.0 mL of NaOH 2M) and 3,5-dinitrosalicylic acid solution (96 mM). Natural and clarified juices (tested at concentrations of 5–1000 µg/mL) and the control were added to the starch solution and left to react with the enzyme for 5 min at 25 °C. The generation of maltose was calculated by the reduction of 3,5-dinitrosalicylic acid to 3-amino-5-nitrosalicylic acid, detectable at 540 nm.

In the α-glucosidase inhibition test, the following solutions were prepared: enzyme solution, maltose solution, *o*-dianisidine solution, and peroxidase/glucose oxidase system-color reagent solution. In the first step, both juices (5–1000 µg/mL) and control were added to the maltose solution. The reaction was started by adding the enzyme solution. Tubes were left for 30 min at 37 °C. Then, perchloric acid was added to stop the reaction. In the second step, the generation of glucose was quantified by the reduction of *o*-dianisidine. The supernatant of the first step was mixed with *o*-dianisidine and peroxidase/glucose oxidase and was left for 30 min at 37 °C. The absorbance was read at 500 nm. In both tests, acarbose was used as a positive control.

### 2.9. In Vivo Assay

#### 2.9.1. Animals

All animal procedures were approved by the Italian Health Ministry (1150/2015-PR), and the mice were obtained, housed, used, and euthanized according to the local ethics committee. The eighteen male Hsd:ICR (CD-1) mice (26−30 days old; weight 32.33 ± 1.32 g) used in the study were obtained from Envigo, The Netherlands. Mice were housed into polycarbonate cages (3 per cage) on wood chips bedding (Lignocel BK 8−15, J. Rettenmaier & Söhne GmbH + Co. KG, Rosenberg) in a ratio of 6 mice per experimental group. The animals were maintained in an automatic light/dark cycle (light periods of 12 h, lights on 0700 to 1900) and acclimatized to laboratory conditions placed from 7 days prior the start of the trial in an air-conditioned room environmentally controlled (temperature 22.0 ± 1.0 °C; humidity: 40.0% ± 10.0%). The animals had free access to water and were fed ad libitum with standard rodent chow (2018 Global rodent diet, Teklad; CP, 186 g/kg; Fat, 62 g/kg; CF, 35 g/kg; Ash, 53 g/kg; ME 13.0 MJ/kg as fed) for one week before assignment to one of the three experimental groups.

#### 2.9.2. Experimental Design

At 5 weeks of age (day “−7”), the mice were marked with ear punch for recognition and randomly assigned to three treatment groups (control, natural juice, and clarified juice). For 28 days, from day “0” to day “27”, each animal received once daily by oral gavage one of the following substances:
Group Control (Control, *n* = 6): mice received 0.2 mL of water to minimize false positive results.Group Natural Juice (NJ, *n* = 6): mice received 500 mg/kg BW of natural juice dissolved into 0.2 mL of water.Group Clarified Juice (CJ, *n* = 6): mice received 500 mg/kg BW of clarified juice dissolved into 0.2 mL of water.

#### 2.9.3. In-Life Evaluations

During the experimental period, the behavior of mice was observed daily, before and after dosing, for abnormalities in clinical signs. Feed intake and water consumption were recorded for each cage every 7 days. Individual BW was recorded once per week (day 0, 7, 14, 21, and 28 of the experiment).

#### 2.9.4. Laboratory Analyses

At the end of the experiment, all mice were anesthetized with CO_2_ and sacrificed by jugulating after overnight fasting (12–13 h), and fresh blood samples were collected into plastic tubes. The serum was separated by centrifugation at 1500× *g* for 15 min and stored at −20 °C until the analyses were performed. Blood chemistry analyses on serum aliquots were performed by an automatic biochemical analyzer (AMS Autolab, Diamond Diagnostics, Holliston, MA, USA) using reagents from Spinreact (Girona, Spain) in order to determine aspartate amino transferase (AST), alanine aminotransferase (ALT), alkaline phosphatase (ALP), total proteins (TP), albumin (ALB), urea (UREA), creatinin (CREA), lactate dehydrogenase (LDH), creatine-phosphokinase (CPK), cholesterol (CHOL), and triglycerides (TRIG).

Reactive oxygen metabolites (d-ROMs) and biological antioxidant potential (BAP) tests were also performed on serum aliquots using reagents from Diacron International s.r.l. (Grosseto, Italy).

### 2.10. Statistical Analysis

Statistical analyses were performed using GraphPad PRISM, version 8.3.0 for Windows (GraphPad Software, San Diego, CA, USA, www.graphpad.com). BW, ADG, feed intake, and water consumption data were analyzed by a two-way ANOVA for repeated measures with Greenhouse-Geisser correction followed by Tukey’s multiple comparison test in order to evaluate the effects of the main factors “group”, “time”, and their interaction “group × time”. In vitro antioxidant and hypoglycemic activity, as well as blood chemistry and oxidative status parameters, were analyzed by one-way ANOVA followed by Tukey’s multiple comparison test. If not all the assumptions for ANOVA were met, the Kruskal-Wallis nonparametric test followed by Dunn’s multiple comparison test was applied. For all data, significance was declared at *p* < 0.05. Results were expressed as means ± standard deviation (SD) unless indicated otherwise.

## 3. Results

### 3.1. Chemical Profile

Physicochemical properties of natural and clarified juices are shown in Table 1. According to the results, suspended solids were completely removed from the fresh juice. No significant differences were found for total soluble solids, as well as phenols, flavonoids, anthocyanins, and ascorbic acid between NJ and CJ.

### 3.2. In Vitro Antioxidant and Hypoglycaemic Activities

Pomegranate juices were studied for their potential antioxidant activity by using three different in vitro tests (Table 2). Juices showed antioxidant activity in a concentration-dependent manner (data not shown). Although CJ is characterized by lower total phenols, anthocyanins, flavonoids, and ascorbic acid content than NJ, it has proved to be more active than untreated juice in all antioxidant tests. The most interesting activity was evidenced for the CJ in the β-carotene bleaching test with IC_50_ values of 19.7 and 44.1 µg/mL after 30 and 60 min of incubation, in comparison to the NJ (IC_50_ values of 51.5 and 55.7 µg/mL after 30 and 60 min of incubation, respectively). Generally, the DPPH radicals scavenging activity is poor. In fact, IC_50_ values of 734.2 and 782.6 µg/mL were found for clarified and natural juices, respectively, in comparison to the positive control ascorbic acid with an IC_50_ value of 5.0 µg/mL.

The inhibition of enzymes involved in the digestion of carbohydrates, such as α-glucosidase and α-amylase, can result in a reduction of the post-prandial increase of blood glucose. Therefore, it can be an interesting management strategy for the treatment of patients affected by type 2 diabetes. As reported in Table 2, NJ was more active than CJ in inhibiting α-amylase (IC_50_ of 67.1 and 76.6 µg/mL for NJ and CJ, respectively), while α-glucosidase was mainly inhibited by CJ (IC_50_ of 80.1.6 and 68.1 µg/mL for NJ and CJ, respectively).

### 3.3. In-Life Evaluations

The BW of the mice increased along the time (*p* < 0.0001; Figure 1A) but not varied among the groups (*p* = 0.9016). Furthermore, the *P. granatum* juice administration did not influence both the feed intake (*p* = 0.2330; Figure 1B) and the water consumption (*p* = 0.2600; Figure 1C) compared to the control group. Even the time did not affect these variables (*p* = 0.4261 and *p* = 0.0778 for feed intake and water consumption, respectively).

### 3.4. Blood Chemistry Parameters and Oxidative Status

Biochemical parameters and oxidative status of the three experimental groups are showed in Table 3. Among blood chemistry parameters, statistical differences were registered for ALT, AST, UREA, CPK, and TRIG values. In particular, NJ group showed values of ALT, AST, and CPK lower than control, while CJ administration was associated with UREA, CPK, and TRIG values lower in respect to the control group.

Oxidative status of mice was ameliorated by the administration of clarified juice. A d-ROMs reduction (*p* = 0.0187) of 32% and a BAP boosting (*p* = 0.0322) of 23% with respect to the control group was observed in this group after 28 days of treatment.

## 4. Discussion

This work aimed to investigate the in vitro and in vivo effects of natural pomegranate juice in comparison to the juice clarified with HF PVDF membrane. In recent years, the use of micro- and ultra-filtration membranes has been extensively investigated for their effects on sensory properties and industrial production techniques as alternatives to conventional clarification procedures [11,12,13], but few studies have focused on the effects of filtration on the juice beneficial properties. HF PVDF membranes completely removed suspended solids, producing a brilliant red juice, but did not modify pH and soluble solids.

These results are consistent with the estimated pore size of PVDF membranes (0.13 µm) and with data reported by Mirsaeedghazi et al. [24] in the clarification of pomegranate juice with PVDF MF membranes in flat-sheet configuration with pore sizes of 0.22 and 0.45 μm. In addition, very low retention of PVDF membranes (in the range of 3.5–4.0%) towards anthocyanins, flavonoids, and phenols were measured. Similar retention values for total phenolics have been reported by Qin et al. [25] in the MF of kiwifruit juice with fly-ash-based ceramic membranes, having an average pore diameter of 2.13 µm. Herein, PVDF-clarified juice showed interesting in vitro antioxidant activity, as well as beta-glucosidase inhibitory activity, in comparison to the natural juice. These results are in agreement with data from our previous study [18] and from the work of Valero et al. [26] that showed as clarification processes are able to produce positive effects on the antioxidant activity despite a reduction of phenols content of pomegranate juice. Indeed, we can deduce that the higher antioxidant effects of clarified juice could be explained through the removal of the antagonism between antioxidants and other constituents.

Both NJ and CJ do not affect the growth of the young mice, nor the feed and water consumption. Previous results on body weight are often controversial, but positive effects on fat reduction have been shown using the pomegranate juice and its extracts [27]. Patel et al. [28] in Wistar rats treated by oral gavage with pomegranate fruit extract at dose levels up to 600 mg/kg BW for 90 days reported no treatment-related biologically significant effects on BW or ADG, feed, and water consumption. In addition, Cerdá et al. [29] reported significant reduction in feed intake and BW related to high levels of the extract (20%) in the diet. It is well-known that the balance between oxidant/antioxidants is essential for the body, but the virtues of the antioxidant substances have been amplified in the last years without solid scientific basis [30]. In this study, a significant decrease of d-ROMs and increase of BAP were observed in the CJ mice. d-ROMs measure the oxidant level within the blood, while BAP matches the total antioxidant capability of plasma and includes either exogenous (ascorbate, tocopherols, and carotenoids) or endogenous components (protein, glutathione peroxidase, superoxide dismutase, and catalase) involved in the overall reactive oxygen species balance [31]. The clarification of pomegranate juice with the PVDF membrane determined the highest protection on ROS production, according to our results obtained in vivo. This is consistent with phytochemical analysis that showed no significant retention of important antioxidant compounds in the MF juice [18], thus confirming the PVDF membrane as a useful tool for juice clarification with no negative effects on its biological activity. Moreover, the antioxidant activity of CJ seemed to be even higher than the NJ, thus suggesting that the filtration process may retain some compounds that limited such activity.

The health benefits of fruits, such as pomegranate, are mainly attributed to the presence of phenols, vitamins, and carotenoids. Importantly, the in vivo effects of antioxidants depend not only on their concentration in fruits and vegetables but also on their bioaccessibility and bioavailability after ingestion. Many studies have focused on the bioavailability of these compounds after their ingestion [30,32]. In general, the absorption and transport processes of many of the potentially bioactive components of fruits and vegetables are complex and not fully understood; thus, prediction of their bioavailability is problematic [10].

It has been demonstrated that the physical state of the food matrix plays a key role in the release, mass transfer, accessibility, and biochemical stability of many food components [10]. Antioxidants are often located in natural cellular compartments or within assemblies produced during processing. In either case, they need to be released during digestion so that they can be absorbed in the gut [10]. Additionally, dietary fibers can reduce the bioavailability of macronutrients, especially fat, and some minerals and trace elements. Since it was demonstrated that pectin strongly decreased the bioavailability of β-carotene [33], dietary fiber is also suspected to affect the absorption of other carotenoids and probably that of α-tocopherol and polyphenols compounds.

Limited information exists on the bioavailability of dietary lipids and carotenoids entrapped in the food matrix. Soluble dietary fibers in the gut could attenuate the absorption of dietary fats and may therefore inhibit the absorption of carotenoids as lipid-soluble compounds [34,35]. The higher antioxidant effect showed by the CJ with respect to the NJ may be due to factors and substances that influence their bioavailability by acting on the digestive process. Despite the filtration process may reduce the amount of antioxidants, it may also reduce the presence of other substances that may interfere with their bioavailability. Further studies would investigate the relation between filtration and bioavailability. This should be of great importance, since the benefits of membrane filtration in preserving healthy substances may also include bioavailability of healthy compounds other than to improve the technological properties of the product.

The findings from the present study following oral gavage administration of pomegranate juices confirmed that these substances do not cause any adverse effects. The administration of both the juices resulted in a decrease of AST and ALT, statistically significant only for NJ and widely recognized as markers of liver damage. This result agrees with Patel et al. [28], who found a decrease of AST and ALT in male rats treated daily by oral gavage with 600 mg/kg BW of *P. granatum* extract. The therapeutic evaluation of natural products in hepatic disorders has been noticed in the last few years, and various natural and synthetic products have been tested by researchers to decrease hepatic damages [36]. Recently, the pomegranate has attracted all the attention due to its potential function in different metabolic pathways and in the improvement of the antioxidant system related to specific tissues and organs, including the liver. Shishavan et al. [37], administering orally pomegranate peel extracts in a ratio of 500 mg/kg BW for 18 days, found a beneficial effect on antioxidant enzymes in rats’ liver.

A decrease of CPK was also noted in the treated groups. CPK is a well-known plasma marker enzyme of heart damage. Its decrease confirms previous studies about the benefits of *P. granatum* administration as a useful nutritional supplement for the prevention and treatment of heart diseases [38]. Similar considerations could be made for the decrease of UREA and TRIG, but they were statistically evident only for the group treated with CJ. Some authors suggested that pomegranate juice is able, by elevating the antioxidant defense system, to protect the kidney against toxicity, thus having a potential protective effect [39]. Esmaillzadeh et al. [40], administering pomegranate juice to diabetic volunteers for eight weeks, found no effect on serum triglycerides but a significant reduction of cholesterol. By contrast, in our study, both juices did not affect cholesterol, and the decrease of TRIG obtained by the CJ may suggest that other factors may be involved in the bioavailability of beneficial substances, thus affecting their activity in vivo.

## 5. Conclusions

In vivo results suggest beneficial properties of pomegranate juice and showed that filtration with hollow fiber PVDF membranes may enhance such effects, mainly in terms of antioxidant activity. This suggests that the filtration process retains some compounds that may limit the absorption of pomegranate juice, reducing the juice beneficial properties. These results suggest that PVDF filtration may represent a useful tool to obtain technological improvements of pomegranate juice, at least without affecting its beneficial properties. The hypothesis that PVDF filtration may increase the bioavailability of some substances is interesting but needs further studies.

## Figures and Tables

**Figure 1 foods-09-00242-f001:**
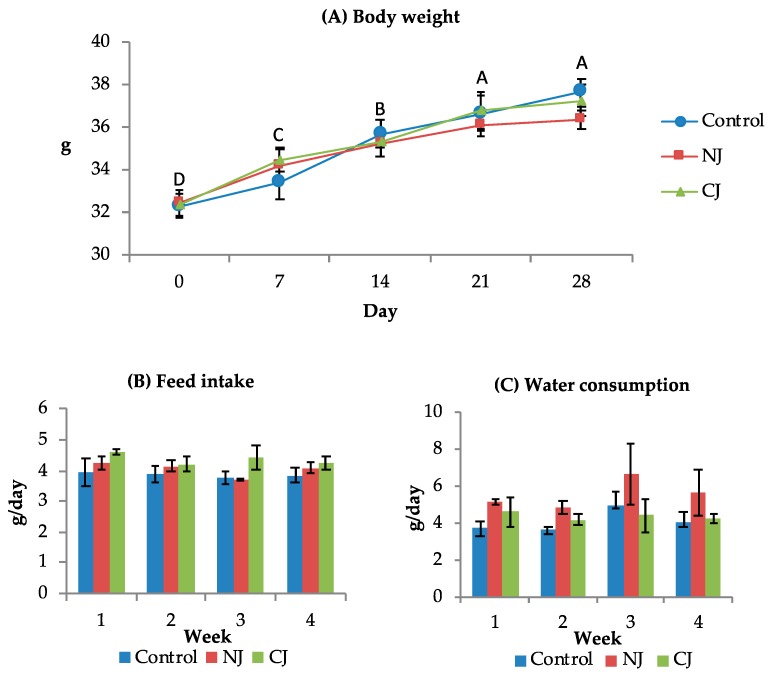
Means ± SEM of the mice body weight (BW) (**A**), feed intake (**B**), and water consumption (**C**). Mice groups received water (control), natural juice (NJ), or clarified juice (CJ) by oral gavage. A–D: different letters indicate significant differences among timepoints *p* < 0.01. No statistically significant interaction Group × Time was observed.

**Table 1 foods-09-00242-t001:** Physicochemical properties of untreated (natural) and clarified pomegranate juice.

Parameter	Natural Juice	Clarified Juice
pH	4.1 ± 0.1	4.0 ± 0.1
Suspended solids (% *w/w*)	4.2 ± 0.1	-
Total soluble solids (°Brix)	22.1 ± 0.4	21.9 ± 0.4
Total phenols ^a^	1989.7 ± 28.3	1919.1 ± 32.3
Total flavonoids ^b^	288.7 ± 7.1	276.8 ± 6.2
Total anthocyanins ^c^	121.3 ± 3.1	116.1 ± 3.5
Ascorbic acid ^d^	132.0 ± 2.5	90.0 ± 1.8

Data are expressed as means ± SD (standard deviation) (*n* = 3). ^a^ mg of chlorogenic acid equivalents/L. ^b^ mg of quercetin equivalents/L. ^c^ mg of cianidin-3-glucoside/L. ^d^ mg/L. -: not found.

**Table 2 foods-09-00242-t002:** In vitro antioxidant and hypoglycemic properties of natural and clarified pomegranate juices.

Sample	Antioxidant Activity				Hypoglycemic Activity	
	DPPH Test (IC_50_ µg/mL)	FRAP Test ^#^ (µM Fe(II)/g)	β-Carotene Bleaching Test (IC_50_ µg/mL)		α-Amylase Inhibitory Assay (IC_50_ µg/mL)	α-Glucosidase Inhibitory Assay (IC_50_ µg/mL)
			30 min	60 min		
NJ	782.6 ± 3.7 ^A^	5.1 ± 1.0 ^C^	51.5 ± 1.9 ^A^	55.7 ± 1.7 ^A^	67.1 ± 2.3 ^B^	80.1 ± 2.8 ^A^
CJ	734.2 ± 2.8 ^B^	14.1 ± 0.6 ^B^	19.7 ± 1.1 ^B^	44.1 ± 1.0 ^B^	76.6 ± 2.2 ^A^	68.6 ± 2.0 ^B^
Positive control *	5.0 ± 0.8 ^C^	63.2 ± 4.5 ^A^	1.0 ± 0.04 ^C^	1.0 ± 0.03 ^C^	50.0 ± 0.9 ^C^	35.5 ± 1.2 ^C^
*P*	<0.0001	<0.0001	<0.0001	<0.0001	<0.0001	<0.0001
*R squared*	1.000	0.9927	0.9975	0.9984	0.9803	0.9918

Data are expressed as means ± SD (*n*= 3). ^#^ at concentration of 2.5 mg/mL. * Positive control: ascorbic acid for DPPH test, BHT for FRAP test, propyl gallate for β-carotene bleaching test, acarbose for α-amylase, and α-glucosidase inhibitory assays. A–C: different letters along the column indicate significant differences *p* < 0.01. NJ: natural juices, CJ: clarified juices, and BHT: butylated hydroxytoluene.

**Table 3 foods-09-00242-t003:** Blood chemistry parameters and oxidative status of the mice after 28 days of trial.

Item		Control		NJ		CJ		*P*	*R squared*
ALT	U/L	61 ± 3.4	A	49 ± 5.9	B	53 ± 8.0	AB	0.0081	0.473
AST	U/L	73 ± 6.6	A	59 ± 5.1	B	67 ± 6.9	AB	0.0058	0.497
ALP *	U/L	90 ± 43.8		158 ± 48.0		134 ± 58.4		0.0802	0.276
TP	g/dL	5.7 ± 0.5		5.7 ± 0.1		6.1 ± 0.3		0.1627	0.231
ALB	mg/dL	3.3 ± 0.3		3.4 ± 0.2		3.3 ± 0.1		0.6976	0.047
UREA	mg/dL	111 ± 6.3	A	91 ± 14.6	A	69 ± 17.8	B	0.0003	0.659
CREA *	mg/dL	0.24 ± 0.1		0.25 ± 0.0		0.26 ± 0.1		0.6925	0.011
LDH	U/L	3458 ± 1250		2094 ± 852		3429 ± 1607		0.1405	0.230
CPK	U/L	341 ± 192	A	80 ± 42	B	91 ± 19	B	0.0017	0.573
CHOL *	mg/dL	210 ± 12		200 ± 14		210 ± 39		0.8026	0.039
TRIG	mg/dL	335 ± 46	a	295 ± 59	ab	248 ± 44	b	0.0279	0.379
d-ROMs	U CARR	151 ± 30	a	149 ± 35	a	102 ± 20	b	0.0187	0.412
BAP	μmol/L	3896 ± 277	b	4607 ± 772	ab	4813 ± 229	a	0.0322	0.434

AST: aspartate amino transferase; ALT: alanine aminotransferase; ALP: alkaline phosphatase; TP: total proteins; ALB: albumin, UREA: urea; CREA: creatinine; LDH: lactate dehydrogenase; CPK: creatine-phosphokinase; CHOL: cholesterol; TRIG: triglycerides; d-ROMs: reactive oxygen metabolites; BAP: biological antioxidant potential. Data are expressed as means ± SD (*n* = 6). Mice groups received by gavage water (control), natural juice (NJ), or clarified juice (CJ). A and B: different letters along the row indicate significant differences *p* < 0.01. a and b: different letters along the row indicate significant differences *p* < 0.05. * data compared by Kruskal-Wallis test followed by Dunn’s multiple comparison test.

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
