# Peer review of "Antioxidant, Biochemical, and In-Life Effects of Punica granatum L. Natural Juice vs. Clarified Juice by Polyvinylidene Fluoride Membrane"

_foods, 2020, doi:10.3390/foods9020242_

Round 1
Reviewer 1 Report
The article ‘Antioxidant, Biochemical, and In-Life Effects of Punica granatum L. Natural Juice vs. Clarified Juice by Polyvinylidene Fluoride Membrane’, presented for review is very interesting. However, some issues need to be clarified or supplemented. The comments are included below.
Line 92 - The authors suggest that few data are reported in the literature on the use of P. granatum juice in healthy mice. For this reason, the choice of dosage to be administered was based on a quantity acceptable to drink daily for a medium-term period and corresponding to about 25 ml of juice for a person of 50 kg of BW. Later, however, they write something different. (Line 362) ‘Patel et al. [27] in Wistar rats treated by oral gavage with pomegranate fruit extract at dose levels up to 600 mg/kg BW for 90 days reported no treatment-related biologically significant effects on BW or ADG, feed and water consumption.’ Please explain the differences. Wouldn't it be advisable to use higher doses of juice in the future? What is the normal amount of juice consumed by a person?
Line 116, 135 - ‘A part of the obtained juice was stored (Natural Juice) at – 17 °C, the other part was clarified (Clarified Juice) by using the preparedPVDF membrane’, ‘The clarified juice was stored at ̶17 °C until analysis.’ - Why was this temperature used? It is neither room temperature nor refrigeration temperature.
Line 140 - ‘The soluble solids content was measured with a refractometer (Atago Co., Ltd, Kumamoto, Japan) at 25 °C. Why was this temperature -chosen?
Line 148, 153, 158, 159 – Why have some of the results been expressed in mg / L and others in mg / 100 ml of juice.
Table 2 - Why FRAP expressed per g and not per ml?
Do the authors know of any other studies showing that despite the lower content of bioactive ingredients, antioxidant activity increases or remains at a similar level? Although clarified juice is characterized by a lower total phenols, anthocyanins, flavonoids, and ascorbic acid content than natural juice, it has proved to be more active than untreated juice in all antioxidant tests. So far, it has been emphasized that filtration always reduces biological activity, and thus deteriorates antioxidant properties. Please search for confirmation of the presented hypothesis.
As the authors write in the conclusions, the hypothesis that PVDF filtration may increase the bioavailability of some substances is quite interesting but requires further studies.
Author Response
Dear Sir,
attached you can find the answers to your queries.
Kind regards

Reviewer 2 Report
The manuscript "Antioxidant, Biochemical, and In-Life Effects of Punica granatum L. Natural Juice vs. Clarified Juice by Polyvinylidene Fluoride Membrane" by Valeria Maria Morittu et. al., describes the use of clarification by polyvinylidene fluoride (PVDF) hollow fiber membrane to ameliorate technological quality of punica granatum L. (pomegranate) juice through the preservation of its bioactive compounds.
EVALUATION
The paper fit the aims and scope of the Foods journal, and the development of clarification processes based on fiber membrane for the improvement of technological quality and maintenance of health benefits in fruit juices processing have scientific interest. Although the followed methodology appears to be appropriate and properly conducted, serious corrections and improvements are needed in this paper to be accepted for publication.
The paper contains several errors and confusing sentences throughout the all text, mainly in Introduction section. The proceedings used in statistical analysis should be better described in Materials and Methods section, where must be explained the selection criteria of the statistical methods.
The Results section should be further explored, where results obtained must be substantiated and properly illustrated, being that the figures of paper can be significantly improved. In addition, some results are scientifically inconsistent. Additionally, the data from the various analysis described in the paper should be provided as supplementary material.
In Discussion section, and despite the results obtained in this article being compared with results previously obtained by other authors, these should be scientifically substantiated and technically explored in order to understand its importance for the improvement of the juice processing industry.
The conclusions reached with results obtained in this paper are overvalued, and this paper should be seen as a preliminary study. The small number of samples and tests, as well as the poor statistical validation of the results obtained, does not allow to confirm the statements contained in the Conclusion section.
Finally, the writing of the article should be deeply reformulated, and English should be greatly improved.
SOME COMMENTS
ABSTRACT
Some abbreviations used in abstract need to be previously described in the text.
INTRODUCTION
Line 40: This sentence is confused. Revise and correct it: “In Italy, owing to the good prospects of the market, pomegranate cultivation has increased considerably in the last 5 years, growing from 50 ha in 2012 to 1,500 ha in 2017, mainly in south of Italy, (Sicily, Puglia, Calabria and Campania regions), with a current total production of about 6,000 tons/year”.
Line 50: This sentence is confusing and out of context, should be reformulated: “At fixed concentrations (5 mg/mL), the extract had the potential to promote for the longevity, formation of new generations, fertility of new generations, and growth properties of Caenorhabditis elegans although higher concentrations significantly reduced these parameters”.
Line 62: This sentence is confused. Revise and correct it: “By contrast, large particles in pomegranate juice are responsible of turbid appearance, which causes some problems in the concentration process, such as off flavours due to the large particles burning on the evaporator walls.”
MATERIAL AND METHODS
Line 243: This text is strongly confused, should be improved. The selection criteria of statistical methods must be described.
RESULTS
The results should be better explored and described. Some results presented scientific inconsistencies. For example, in Table 3, the Standard Deviation (SD) values are too high. The error between analysis should be presented in Relative Standard Deviation (% RSD). The statistical analysis must be strongly improved. Revise and confirm it.
DISCUSSION
The discussion section should be better elaborated, the results should be scientifically reviewed and technically explored, and if possible, the results obtained should be compared with similar results from the previously published scientific papers. Revise and correct it.
CONCLUSIONS
The conclusions reached with results obtained in this paper are overvalued. For example: Line 431: “In vivo results confirmed the beneficial properties of pomegranate juice and showed that filtration with hollow fiber PVDF membrane may enhance such effects mainly in terms of antioxidant activity.”

Author Response

(The authors gave the same response as above.)

Reviewer 3 Report
In this study, the authors evaluated the in vitro antioxidant and hypoglycaemic activity of natural pomegranate and clarified (PVDF) juice on mice. Although the work has some interesting points, there are some important aspects that should be improved before its publication:
- In the abstract, please not use multiple abbreviations.
- There are several typographical errors, the italic letter should be used for in vivo/in vitro, ad libitum, etc. Please, change “hearth” diseases by heart diseases (lines 416 and 417).
- Line 50. Please, specify on worms because, in this form, the sentence has no meaning.
- When you use abbreviations the first time it should appear in the rest of the main text.
- Why did not use two types of clarified juices (PVDF and PSU) in comparison with natural juice? It would provide novelty to your research due to in the last decade there are several studies about clarified juice of pomegranate on human health.
- In the juice preparation, do you add antioxidants to avoid browning? Could it affect the antioxidant activity?
- Line 132. Regarding the enzymatic solution, what enzymes did it contain?
- The clarified juice was stored at -17 °C until analysis, Could the freeze-defrost cycle affect the chemical profile?
-The fact that although clarified juice had a lower bioactive compound profile than natural juice, the clarified juice was more active in all antioxidant tests. It should be discussed in more depth.
- Regarding the hypoglycaemic activity, both NJ and CJ had effects. In this context, another type of clarification (e.g.; PSU) would be very useful.
-Lines 386-393 correspond to human data. It is not fit in the current study on mice.
-Line 400. Please, add a new reference to the sentence: “it may also reduce the presence of other substances that may interfere with their bioavailability”.
-Line 410. Please, add a new reference to the phrase: “various natural and synthetic products have been tested by researchers to decrease hepatic damages”.
- Lines 418-421. This paragraph belongs to the results section. It should be removed from the discussion.
- In the conclusion section, the phrase “the filtration process, thus increasing the bioavailability of functional compounds and, as a consequence, their beneficial effects” is too speculative. It should be rewritten.
- Please, remove the horizontal lines of all figures for improving the understanding.
Author Response

(The authors gave the same response as above.)

Round 2
Reviewer 1 Report
Comments submitted to the authors have been taken into account. The article in its current form is suitable for printing.
Reviewer 2 Report
The changes intended / suggested by the reviewer were introduced in the new version of the paper. The current version is suitable for publication.
Reviewer 3 Report
All issues concerning this manuscript were solved.